# Risk Factors for the Spread of Brucellosis in Sheep and Goats in the Campania Region in the Years 2015–2020

**DOI:** 10.3390/microorganisms11112623

**Published:** 2023-10-24

**Authors:** Roberta Brunetti, Maria Ottaiano, Mario Fordellone, Paolo Chiodini, Simona Signoriello, Federica Gargano, Fabrizio De Massis, Loredana Baldi, Esterina De Carlo

**Affiliations:** 1Istituto Zooprofilattico Sperimentale del Mezzogiorno, 80055 Portici, Italy; roberta.brunetti@izsmportici.it (R.B.); federica.gargano@izsmportici.it (F.G.); loredana.baldi@izsmportici.it (L.B.); 2Università degli studi della Campania Luigi Vanvitelli, 80138 Naples, Italy; mario.fordellone@unicampania.it (M.F.); paolo.chiodini@unicampania.it (P.C.); simona.signoriello@unicampania.it (S.S.); 3Reference Laboratory for Brucellosis (Brucella abortus, Brucella melitensis, Brucella suis), Reference Laboratory for Ovine Epididymitis (Brucella ovis), Istituto Zooprofilattico Sperimentale dell’Abruzzo e del Molise “G. Caporale”, 64100 Teramo, Italy; f.demassis@izs.it; 4National Reference Centre for Hygiene and Technologies of Water Buffalo Farming and Productions, Istituto Zooprofilattico Sperimentale del Mezzogiorno, 80055 Portici, Italy; esterina.decarlo@izsmportici.it

**Keywords:** brucellosis, risk factors, regression model, statistical association, sheep and goats

## Abstract

Brucella is a Gram-negative facultative intracellular pathogen that causes infection in sheep and goats (*B. melitensis.*); *B. melitensis* can also infect other animals. Sheep and goat brucellosis is still present in some regions of Italy, including Campania, and causes considerable economic losses and health threats. The aim of this study was to evaluate the possible risk factors influencing the spread of brucellosis among sheep and goat farms in the Campania region in order to provide the local veterinary services with practical support in evaluating and planning diagnostic, preventive and control interventions. The results of official controls for brucellosis carried out from 2015 to 2020 in the sheep and goat farms of the Campania Region were analyzed. Data were extracted from the National Veterinary Information Systems and the Laboratory Management System of the Experimental Zooprophylactic Institute of Southern Italy. Statistical analysis was carried out through the software R version 4.1.0; the dataset consisted of 37,442 observations, and 9 qualitative and quantitative variables were evaluated on 8487 farms, 248 of which were positive. The association between covariates and the outcome (presence/absence of the disease) was evaluated (Fisher and Wilcoxon tests). A logistic regression model with mixed effects was carried out. This study confirmed that brucellosis in sheep and goats in the Campania region mostly occurs through contact with infected animals imported from other farms (OR = 3.41—IC 95% [1.82–6.41]). Farms with a greater number of animals were seen to be at the greatest risk of infection (OR = 1.04—IC 95% [1.03–1.05]); previous suspension of healthy status also proved to be a risk factor (OR = 55.8—IC 95% [26.7–117]).

## 1. Introduction

Brucellosis is one of the world’s most significant zoonosis and is caused by infection with members of the genus Brucella (McGiven J., 2014 [1]). Brucellosis remains the most important zoonotic disease, affecting humans and livestock worldwide (Laine et al., 2022 [2]; Moreno et al., 2022 [3]). Brucellosis is a zoonotic disease that has serious animal welfare and economic consequences worldwide (Elrashedy, A et al., 2022 [4]). Ovine and caprine brucellosis, caused by the bacterium *Brucella melitensis*, is one of the world’s most widespread zoonoses and is a major cause of economic losses in domestic ruminant production (Spink et al., 1962 [5]; Whatmore et al., 2009 [6]; Corbel, 1997 [7]; Socorro Ruiz-Palma MD et al., 2021 [8]). *B. melitensis* is also the most common cause of brucellosis in humans. While small ruminants are the natural host, *B. melitensis* can also infect other animals, such as cattle (*Bos taurus*), water buffalo (*Bubalus bubalis*) and camels (*Camelus* spp.) (WOAH, 2016 [9]; Refai, 2022 [10]; Di Giannatale et al., 2008 [11]; De Massis et al., 2005 [12]; John McGiven, 2014 [1].

The infection of a flock/herd causes economic losses due to abortions (which occur between the 3rd and 4th months of pregnancy, with possible retention of the fetal membranes) and decreased milk production (Calistri et al., 2013 [13]). Shedding of *B. melitensis* through uterovaginal secretion following an abortion or infectious lambing lasts longer than in cows infected with *B. abortus*; in infected goats, especially after abortion, the localization of *B. melitensis* in the mammary gland markedly reduces milk secretion, an effect which can last for the entire period of lactation. Airborne and venereal infections are established more frequently in sheep and goats than in cattle (Rossetti CA et al., 2017 [14]). Sheep show greater resistance to infection than goats (Nielsen and Duncan, 1990 [15]). *Brucellae* are eliminated mainly through milk, abortions and uterine excretions and, to a lesser extent, through several other secretions and excretions. The environment is a possible source of contagion. In favorable conditions, the bacteria can survive for a long time outside the host, and the disease can spread through the contamination of water, foodstuffs and soil (Nielsen and Duncan, 1990 [15]). Animals such as cats and dogs can become infected by ingesting milk from infected farms (G. Wareth et al., 2017 [16]).

Humans can contract the disease through contact with biological material or infected animals, via aerosols (a high risk for professional categories such as laboratory workers) or through the ingestion of contaminated products of animal origin. Human brucellosis is a systemic infectious disease with varying clinical manifestations (De Massis et al., 2019 [17]). Patients often develop a fever of unknown origin with an insidious clinical onset (De Massis et al., 2019 [17]), (Pappas et al., 2006 [18], WHO, 68 2006 [19]). The disease is often difficult to diagnose because of its similarities to other febrile diseases, such as malaria or other undulating fevers, and it occurs as a subacute or chronic illness that is generally not lethal [(De Massis et al., 2019 [17])]. The acute stage is characterized by nonspecific symptoms similar to those of a flu-like or septicemic illness. Clinical manifestations may be the effect of many disorders, such as osteoarticular, dermal, gastrointestinal, respiratory, cardiovascular and neurological involvement, thus mimicking many other infectious and noninfectious diseases (Shirima GM—Tanz J Hlth Res (2010) [20]; Soares de Araujo Teixeira et al., 2017 [21]). Direct invasion of the central nervous system may occur in about 5% of cases (*B. melitensis*), and meningitis or meningoencephalitis is the most common finding. *Brucella* spp. meningitis can be acute or chronic. Although it often occurs late in the course of the disease, it may also be the presenting manifestation (Pappas et al., 2006 [18], WHO, 68 2006 [19]). However, although their occurrence is rare, endocarditis and neurobrucellosis may be fatal. (Pappas et al. [18], 2006, WHO, 2006 [19]). Seasonal peaks in the number of cases in humans have been described in the literature and have been correlated with the sheep and goat lambing season (De Massis et al., 2005 [17]). The European countries most affected by brucellosis are those of the Mediterranean; indeed, in 2008, approximately 85% of reported human brucellosis cases occurred in Greece, Italy, Portugal and Spain (genus Brucella. (2010) http://www.bacterio.net/brucella.html (accessed on 8 February 2016) [22]—De Massis et al., 2019 [17]). The cases reported in Northern European countries are mainly “imported cases”, as they are associated with people returning from travel to countries where brucellosis is endemic. In 2021, 162 confirmed cases were reported in the EU, which was a slight increase from 2020. The notification rate was 0.03 cases per 100,000 populations. (EFSA, 2010–EFSA 2022 [23]).

In both animals and humans, brucellosis can be diagnosed by means of the conventional culture method, various serological tests and molecular techniques (Radostits OM et al. [24]).

Factors favoring the persistence of the bacterium are the lack of an adequate surveillance system, the high density of animals, close contact between different susceptible species, and the poor management and low level of biosecurity of farms (Kabagambe et al., 2001 [25]; M. Dadar et al., 2019 [26]). The main risk factors include the introduction of an infected animal into a healthy population, incorrectly managed abortions, the use of contaminated milk, drinking water or food and poor veterinary practices (use of contaminated tools) (Huan Zhang et al., 2020 [27]). 

At a national level, brucellosis is still present. Although most of the provinces of northern and central Italy have acquired the status of “free territory”, the disease persists in the regions of southern Italy and in Sicily (Reg. UE 2023/1071—1 June 2023 [28]). The national eradication plan provides for periodic serological testing on cattle and buffalo farms and on sheep flocks and goat herds; the interval between testing and the number of farms and animals to be tested varies according to the health status of the province or region concerned. Specifically, in nonfree provinces, the Ministry of Health issued a ministerial order whereby the eradication measures were intensified, and stricter provisions were issued for the detection and slaughter of infected animals (FAO Ministerial Ordinance May 2015: Extraordinary veterinary police measures regarding tuberculosis, bovine and buffalo brucellosis, sheep and goat brucellosis, enzootic bovine leucosis [29]).

All official tests are performed at the Experimental Zooprophylactic Institutes responsible for the specific area. By the Ministerial Decree of 4 October 1999, the National Reference Center for Brucellosis was activated at the headquarters of the G. Caporale Institute. One of its tasks is to confirm, whenever required, the diagnosis of brucellosis made by other laboratories.

In the Campania region, infection displayed a decreasing trend from 2015 to 2017; in recent years, the disease has still been present in the region, albeit with reduced prevalence levels. In 2018 and 2020, the incidence and prevalence revealed the same value. In 2019, the prevalence (0.16%) and the incidence (0.13%) were almost the same value (national reporting information system SIR—Vetinfo, https://www.vetinfo.it/j6_rendicontazioniNew/report/ZOB/ (accessed on 15 April 2021) [30]). This suggests that a constant source of infection persists, generating new outbreaks every year. About 50% of the new positive farms (4 out of a total of 10 positive farms) were detected in the Province of Salerno, where, for years, a 100% rate of examinations had never been achieved. The aim of this study was to evaluate the possible risk factors influencing the spread of brucellosis on sheep and goat farms in the Campania region, in order to provide the local veterinary services with practical support in evaluating and planning diagnostic, preventive and control interventions.

## 2. Materials and Methods

A longitudinal observational study was conducted on cohorts of sheep and goats: the flocks/herds tested differed only in terms of exposure to possible risk factors. This retrospective study involved the analysis of a 6-year period (from 2015 to 2020) and was based on data on all sheep and goat farms in the Campania region in the period considered.

The data were extracted from the National Veterinary Information Systems (NDb) and the Laboratory Management System (SIGLA) of the Experimental Zooprophylactic Institute of Southern Italy. The data provided by SIGLA were used to identify the flock/herd, its outcome (positive/negative) and the year. The national database of animal identification and registration (NDb) includes data on all sheep and goat farms nationwide and all movements of each animal during its life (https://www.vetinfo.it/ (accessed on 15 April 2021)). The results of official controls carried out from 2015 to 2020 in the sheep and goat farms of the Campania Region were analyzed. In accordance with the legislation in force, animals were tested by means of the rapid serum agglutination test (RBT) and/or complement fixation test (CFT), the official serological tests considered.

The flock/herd was defined as “infected” (positive) when one or more of its animals proved positive on serological tests in the period considered. 

Inclusion Criteria:Presence of sheep/goats on the farm;Farm located in the Campania Region;Farm open at least one day in the period considered, 2015–2020;Farm controlled for brucellosis in the years 2015–2020.Exclusion criteria:Farm without sheep or goats;Farm not controlled for brucellosis in the years 2015–2020.

For each year, the intrafarm prevalence was calculated as the number of positive animals on the farm in proportion to the number of animals tested.

Statistical analysis was carried out by means of the software R version 4.1.0. (R Foundation for Statistical Computing, Vienna, Austria). Continuous variables were reported as medians and interquartile ranges (IQRs). Categorical variables were reported as percentages. 

A preliminary analysis was conducted only on positive farms by calculating the percentage of recurrence of positives: out of 248 positive farms, only 26 suffered repeated outbreaks (12%). Being less than 50%, this percentage allowed the statistical analysis to be carried out.

The analyses were carried out for each single year: the possible dependence between each single covariate and the outcome was evaluated. For qualitative variables, Fisher’s exact test was used, as the sample size per year was small, especially for positive farms. For quantitative variables, the nonparametric Wilcoxon test was used. A *p*-value < 0.05 was considered significant.

The data recorded in the dataset constitute the information on the sheep and goat farms tested for Brucellosis from 2015 to 2020 in the Campania region. In order to be able to consider the time variable and, therefore, to evaluate the effect of the single covariates on the outcome over time, we constructed a logistic regression model with mixed effects.

Mixed effects logistic regression is used to model binary outcome variables, the logistic probabilities of outcomes being modeled as a linear combination of the predictor variables when the data are pooled or when there are both fixed and random effects. A mixed model is a statistical model containing both fixed effects and random effects. 

The response variable is then modeled by combining fixed effects, which are common to the whole population, with random effects, which vary among individuals.

The model had the positive/negative outcome of the individual farm as the response variable, the year variable as the fixed effect and the individual farm as the random effect. On the basis of the preliminary analyses, only those variables that displayed an association with the outcome were entered into the model. As the number of movements and displacements were correlated, we decided to insert only the variable “movements” into the model as an explanatory variable, as this was seen to be statistically associated with the outcome during the preliminary analysis. Before being included in the model, the Yes/No dichotomous variables were transformed into “factors” by associating them with the presence of the covariate “1” and the absence of “0”, respectively.

The significance of the variables within the final model was confirmed by means of the ANOVA test. With a *p*-value < 0.05, the ANOVA of the model confirms that the explanatory variables are significant and well predict the response variable.

## 3. Results

The percentage of positive farms with repeated outbreaks over the years was calculated: 26 out of 248 farms suffered repeated outbreaks. The dataset consisted of 37,442 observations; 9 qualitative and quantitative variables were measured on 8487 farms, 248 of which were positive according to the NDb and SIGLA data (Table 1, Table 2, Table 3, Table 4, Table 5 and Table 6: frequency table, by year). Most of the farms in the period considered had an intrafarm prevalence below 0.1.

Figure 1 shows the box plots of the variable number of animals on the farms, divided into positive and negative farms: as can be seen from the graphs, the positive farms are those with a greater number of animals.

A significant association emerged between the presence of infection and the productive orientation of the farm. Moreover, a statistically significant association was also found between the presence of infection and the province. Indeed, most of the positive farms in the period considered were situated in the province of Salerno. A statistically significant association was found between animal movements and the study outcome (presence/absence of infection). The highest average number of movements (3.9) for positive farms was found in the year 2019. In addition, a statistically significant association was seen between the presence of infection and the average number of animals on the farm, with the disease occurring more frequently on farms with a higher number of animals. Indeed, in 2019, the infection was detected on farms with an average of 193 animals. Finally, a statistically significant association was observed between the presence of infection and the previous suspension of the farm’s healthy status; in 2019, 73.5% of farms that had previously had their healthy status suspended owing to brucellosis tested positive for the disease (Table 1, Table 2, Table 3, Table 4, Table 5 and Table 6).

No statistically significant association was found between the variable “pasture” and the response variable (*p*-value > 0.05). Few grazing herds had the disease; in the years 2018 and 2019, no farm on which animals were turned out to pasture tested positive for brucellosis (Table 1, Table 2, Table 3, Table 4, Table 5 and Table 6).

There was no association between the presence of cattle and the study response variable (Table 1, Table 2, Table 3, Table 4, Table 5 and Table 6).

For each covariate, single univariate logistic models with mixed effects were created in order to evaluate the trend over time of each risk factor. As the covariates “number of movements” and “movements” were correlated, we inserted only the variable “movements” in the model as an explanatory variable. On the basis of the preliminary analyses, only those variables that showed an association were inserted into the model. After evaluating the explanatory variables and the iterations with the variable “year”, we carried out the additive logistic regression model with mixed effects by inserting the three variables: “Movements”, “Suspension of qualification” and “Number of animals in the herd”. Regarding the variable “Number of animals in the herd”, the relationship between each 10-unit increase in the number of animals on the farm and the study outcome was evaluated (Table 7).

The OR of the variable “movements” was 3.41, indicating that it is a risk factor for brucellosis; farms that moved animals had a 3.41 times higher risk of contracting the disease than those without movements. On analyzing the iteration of movements over time, it can be seen that there was a fluctuating trend over the years. The OR was statistically significant for the years 2016, 2018 and 2019, increasing in 2019 to 9.22 (Table 7).

Concerning the “Suspension of healthy status” variable, it can be seen that farms that had previously had their healthy status suspended had a higher risk of developing the disease, with an OR of 55.8. On iterating over time, the OR proved significant from 2018 to 2020, and, above all, in 2019, the OR was 34.6. Thus, farms that had had their healthy status suspended had a 34.6 times higher risk of contracting the disease than farms that had never undergone suspension of their healthy status (Table 7).

Regarding the variable “number of animals on the farm”, every increase of 10 animals in the herd led to a 4% higher risk of contracting the disease. Indeed, since brucellosis is mainly transmitted by direct contact, the presence of a greater number of animals in the herd increases the risk of spreading it.

## 4. Discussion

The data used in this study were not collected specifically to identify risk factors for the onset and/or persistence of brucellosis in sheep and goats. Rather, they were collected on a regular basis with the main purpose of monitoring the activities and results of the eradication program. Using data collected ad hoc would have allowed a broader and more specific view but would have been considerably more costly. Furthermore, for obvious economic reasons, an ad hoc study cannot be carried out as a census of the entire animal population of a region. Thus, the data used in our study were more representative than those provided by an ad hoc study (Calistri et al., 2013 [13]).

The aim of recording all control activities is to monitor the progress of the disease-prevention plan and the achievement of its objectives. Moreover, the availability of this large amount of data is a prerequisite to adequate epidemiological analyses, which are fundamental to the correct rescheduling of activities based on risk analysis. Indeed, in the final phase of an eradication program, it is essential to identify possible risk factors that affect the spread and maintenance of infection in the animal population and to search for any residual sources of infection. This is necessary in order to improve control and prevention (Nannini et al.,1992 [31]).

Such data are particularly useful in contexts in which repeated measurements are made on the same statistical units (longitudinal study) or in which measurements are made on clusters of related statistical units. Because of their ability to address missing values, mixed-effects models are often preferred over more traditional approaches, such as the repeated-measures analysis of variance. Furthermore, the mixed-effects model (Ricci, V. (2006) [32]) allows some parameters to vary randomly so as to take into account natural heterogeneity in the population.

The present study confirmed that brucellosis in sheep and goats in the Campania Region mostly occurs through contact with infected animals imported from other farms. Indeed, both farms that introduced potentially infected animals from other farms and those with a greater number of animals were seen to have a higher risk of harboring infection. In this regard, it should be clarified that premove tests are carried out, but quarantine is not. Indeed, farms implement almost no corporate management measures and no structural biosecurity measures. Unfortunately, little weight has been given to this in the past.

Regarding the absence of an association between the variable “pasture” and the outcome of this study, it should be borne in mind that sometimes the movement of livestock to pasture is not tracked by veterinary services through information systems. Furthermore, it should also be considered that since the entire sheep and goat livestock population has not been checked over the years, the infection may not have been detected, thus invalidating the result of the absence of association. Another aspect to consider is that not all pastures are georeferenced on Vetinfo, and, in any case, during the processing phase, georeferencing includes only two coordinates (latitude and longitude). Some of the territories most affected by the disease, such as the province of Salerno, have large areas of land subject to the transhumance of both cattle and sheep/goats; this favors the direct and indirect contact of animals with infectious diseases. In this regard, the lack, or incomplete traceability, of such information does not allow us to estimate the risk; thus, the result obtained, i.e., the absence of association between the presence of cattle on the farm and the outcome, as in the case of pastures, could be misleading. It would, therefore, be desirable to carry out a subsequent study of the territory destined for grazing and to trace all the animals that have crossed it in order to quantify the risk of infections connected with such movements. Indeed, our analyses revealed that the disease was more frequent in the province of Salerno (Fisher test with *p*-value < 0.05%); in this regard, it would be useful to investigate the possible association/correlation between brucellosis in sheep, goats and cattle and the widespread practice of the transhumance of cattle in this province, a practice which, in the past, hindered the containment of infection in the area.

Abortion is the most predominant symptom of brucellosis in naturally infected sheep. The animals commonly abort only once, but reinvasion of the uterus and the shedding of organisms can occur during subsequent pregnancies. Some infected animals carry the pregnancy to term and shed the organism. It can, therefore, be concluded that abortion in infected animals impacts public health (Benkirane A. et al. [33]). The absence of a statistically significant association between abortions and the presence of the infection could be attributable to the failure to notify abortions. Indeed, breeders often attach little importance to this event and do not inform the Veterinary Services (S.V.) of the AA.SS.LL. Moreover, no herd today has an agricultural veterinarian who can notify such events. In this regard, it is necessary to raise the awareness of all stakeholders regarding the importance of notifying abortions on farms so that the competent authorities can carry out all the necessary examinations to exclude and/or confirm infection and ensure that all precautionary measures are implemented in order to avoid any spread of the pathogen inside and outside the farm. Moreover, in EU REG 429/2016 [34], which came into force on 21 April 2021, this concept reigns supreme.

The present study demonstrates the importance of the availability of detailed and reliable epidemiological data. Indeed, the implementation of efficient information systems is a fundamental prerequisite to evaluating and replanning veterinary activities, thereby enabling continuous monitoring of the health status of animal populations (Calistri et al., 2010 [13]).

Finally, we observed that farms that had had their healthy status suspended had a greater probability than the others of developing the disease over the years. Although unsurprising, this finding should urge the competent authority to implement timely and targeted epidemiological investigations in all farms as soon as infection is detected. This could avoid the spread of the pathogen to the whole farm and mitigate the risk of spreading the infection to other farms and the external environment. It is, therefore, necessary to sensitize all stakeholders and make the Veterinary Services of the AA.SS.LL. aware of their role in eradicating the disease.

## 5. Conclusions

In conclusion, brucellosis in sheep and goats is still a problem in some regions, including Campania. Over the years, numerous funds have been allocated to the eradication of this infection, though with scant results. Our study focused on the critical issues and risk factors responsible for the presence, albeit limited, of brucellosis in sheep and goats in the Campania region. This analysis provides tools that can enable resources to be concentrated where necessary, without effort or waste. The fact that farms that had previously had their qualifications suspended were at higher risk of infection highlights the need to improve the management of livestock farms, especially in the case of large enterprises. To date, this aspect has been underestimated. Our study provides concrete evidence of what has already been reported in the literature.

## Figures and Tables

**Figure 1 microorganisms-11-02623-f001:**
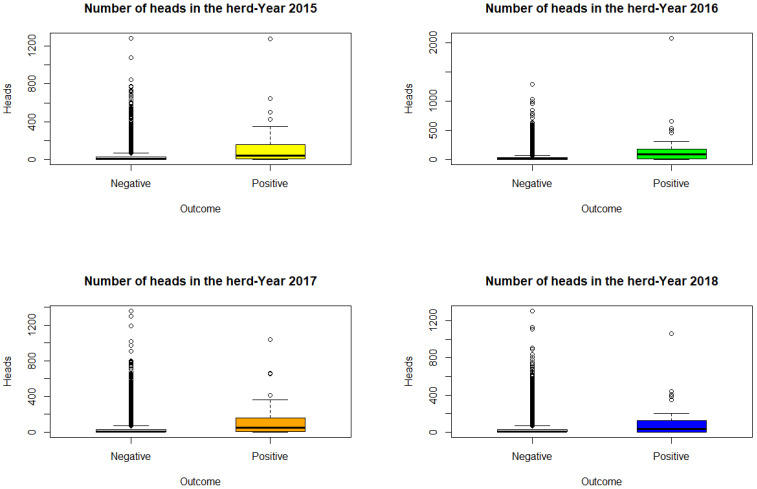
Box Plot of n. of animals in the herd, by year.

**Table 1 microorganisms-11-02623-t001:** Fisher and Wilcoxon tests and frequency table (2015).

	Negative	Positive	*p*-Value
	(N = 6163)	(N = 68)	
**Province**			
Avellino	1623 (26.3%)	9 (13.2%)	<0.05
Benevento	1338 (21.7%)	7 (10.3%)	
Caserta	715 (11.6%)	4 (5.9%)	
Napoli	440 (7.1%)	0 (0%)	
Salerno	2047 (33.2%)	48 (70.6%)	
**Productive orientation**			
Meat	2790 (45.3%)	24 (35.3%)	<0.05
Milk/Wool	65 (1.1%)	3 (4.4%)	
Mixed	1764 (28.6%)	24 (35.3%)	
Multi-production	280 (4.5%)	9 (13.2%)	
Self-consumption	1264 (20.5%)	8 (11.8%)	
**Presence of cattle in farm**			
No	3520 (57.1%)	38 (55.9%)	0.902
Yes	2643 (42.9%)	30 (44.1%)	
**Abortions**			
No	6157 (99.9%)	66 (97.1%)	<0.05
Yes	6 (0.1%)	2 (2.9%)	
**Number of animals on the farm**			
Mean (SD)	37.0 (77.7)	119 (195)	<0.05
Median [Min, Max]	10.0 [0, 1280]	40.0 [0, 1280]	
**Pasture**			
No	6091 (98.8%)	65 (95.6%)	<0.05
Yes	72 (1.2%)	3 (4.4%)	
**Movements**			
No	5274 (85.6%)	50 (73.5%)	<0.05
Yes	889 (14.4%)	18 (26.5%)	
**Number of Movements**			
Mean (SD)	0.197 (0.576)	0.441 (0.853)	<0.05
Median [Min, Max]	0 [0, 8.00]	0 [0, 3.00]	
**Suspension of the healthy status**			
No	6010 (97.5%)	43 (63.2%)	<0.05
Yes	153 (2.5%)	25 (36.8%)	

**Table 2 microorganisms-11-02623-t002:** Fisher and Wilcoxon tests and frequency table (2016).

	Negative	Positive	*p*-Value
	(N = 6201)	(N = 41)	
**Province**			
Avellino	1577 (25.4%)	9 (22.0%)	<0.05
Benevento	1320 (21.3%)	2 (4.9%)	
Caserta	724 (11.7%)	4 (9.8%)	
Napoli	424 (6.8%)	2 (4.9%)	
Salerno	2156 (34.8%)	24 (58.5%)	
**Productive orientation**			
Meat	2862 (46.2%)	16 (39.0%)	<0.05
Milk/Wool	63 (1.0%)	2 (4.9%)	
Mixed	1748 (28.2%)	15 (36.6%)	
Multi-production	276 (4.5%)	5 (12.2%)	
Self-consumption	1252 (20.2%)	3 (7.3%)	
**Presence of cattle in farm**			
No	3524 (56.8%)	29 (70.7%)	0.0822
Yes	2677 (43.2%)	12 (29.3%)	
**Abortions**			
No	6193 (99.9%)	41 (100%)	1
Yes	8 (0.1%)	0 (0%)	
**Number of animals on the farm**			
Mean (SD)	39.0 (81.8)	179 (345)	<0.05
Median [Min, Max]	11.0 [0, 1290]	91.0 [0, 2080]	
**Pasture**			
No	6150 (99.2%)	40 (97.6%)	0.291
Yes	51 (0.8%)	1 (2.4%)	
**Movements**			
No	5366 (86.5%)	33 (80.5%)	0.251
Yes	835 (13.5%)	8 (19.5%)	
**Number of Movements**			
Mean (SD)	0.188 (0.601)	0.415 (1.02)	0.2
Median [Min, Max]	0 [0, 15.0]	0 [0, 4.00]	
**Suspension of the healthy status**			
No	6025 (97.2%)	25 (61.0%)	<0.05
Yes	176 (2.8%)	16 (39.0%)	

**Table 3 microorganisms-11-02623-t003:** Fisher and Wilcoxon tests and frequency table (2017).

	Negative	Positive	*p*-Value
	(N = 6541)	(N = 39)	
**Province**			
Avellino	1536 (23.5%)	4 (10.3%)	<0.05
Benevento	1256 (19.2%)	1 (2.6%)	
Caserta	712 (10.9%)	2 (5.1%)	
Napoli	397 (6.1%)	3 (7.7%)	
Salerno	2640 (40.4%)	29 (74.4%)	
**Productive orientation**			
Meat	2886 (44.1%)	14 (35.9%)	<0.05
Milk/Wool	75 (1.1%)	1 (2.6%)	
Mixed	1754 (26.8%)	14 (35.9%)	
Multi-production	305 (4.7%)	6 (15.4%)	
Self-consumption	1521 (23.3%)	4 (10.3%)	
**Presence of cattle in farm**			
No	3842 (58.7%)	25 (64.1%)	0.52
Yes	2699 (41.3%)	14 (35.9%)	
**Abortions**			
No	6533 (99.9%)	38 (97.4%)	0.0521
Yes	8 (0.1%)	1 (2.6%)	
**Number of animals on the farm**			
Mean (SD)	39.3 (86.2)	148 (223)	<0.05
Median [Min, Max]	10.0 [0, 1360]	53.0 [0, 1040]	
**Pasture**			
No	6498 (99.3%)	39 (100%)	1
Yes	43 (0.7%)	0 (0%)	
**Movements**			
No	5574 (85.2%)	30 (76.9%)	0.171
Yes	967 (14.8%)	9 (23.1%)	
**Number of Movements**			
Mean (SD)	0.208 (0.612)	0.359 (0.811)	0.134
Median [Min, Max]	0 [0, 12.0]	0 [0, 4.00]	
**Suspension of the healthy status**			
No	6464 (98.8%)	22 (56.4%)	<0.05
Yes	77 (1.2%)	17 (43.6%)	

**Table 4 microorganisms-11-02623-t004:** Fisher and Wilcoxon tests and frequency table (2018).

	Negative	Positive	*p*-Value
	(N = 6386)	(N = 40)	
**Province**			
Avellino	1495 (23.4%)	8 (20.0%)	<0.05
Benevento	1199 (18.8%)	3 (7.5%)	
Caserta	702 (11.0%)	1 (2.5%)	
Napoli	394 (6.2%)	0 (0%)	
Salerno	2596 (40.7%)	28 (70.0%)	
**Productive orientation**			
Meat	2785 (43.6%)	21 (52.5%)	<0.05
Milk/Wool	71 (1.1%)	3 (7.5%)	
Mixed	1674 (26.2%)	11 (27.5%)	
Multi-production	307 (4.8%)	0 (0%)	
Self-consumption	1549 (24.3%)	5 (12.5%)	
**Presence of cattle in farm**			
No	3775 (59.1%)	22 (55.0%)	0.63
Yes	2611 (40.9%)	18 (45.0%)	
**Abortions**			
No	6371 (99.8%)	39 (97.5%)	0.0952
Yes	15 (0.2%)	1 (2.5%)	
**Number of animals on the farm**			
Mean (SD)	39.5 (85.9)	107 (195)	<0.05
Median [Min, Max]	10.0 [0, 1310]	38.5 [0, 1060]	
**Pasture**			
No	6311 (98.8%)	40 (100%)	1
Yes	75 (1.2%)	0 (0%)	
**Movements**			
No	5318 (83.3%)	27 (67.5%)	<0.05
Yes	1068 (16.7%)	13 (32.5%)	
**Number of Movements**			
Mean (SD)	0.244 (0.687)	0.750 (1.55)	<0.05
Median [Min, Max]	0 [0, 10.0]	0 [0, 8.00]	
**Suspension of the healthy status**			
No	6348 (99.4%)	24 (60.0%)	<0.05
Yes	38 (0.6%)	16 (40.0%)	

**Table 5 microorganisms-11-02623-t005:** Fisher and Wilcoxon tests and frequency table (2019).

	Aziende Neg	Aziende Pos	*p*-Value
	(N = 6185)	(N = 34)	
**Province**			
Avellino	1455 (23.5%)	2 (5.9%)	<0.05
Benevento	1156 (18.7%)	2 (5.9%)	
Caserta	669 (10.8%)	2 (5.9%)	
Napoli	377 (6.1%)	2 (5.9%)	
Salerno	2528 (40.9%)	26 (76.5%)	
**Productive orientation**			
Meat	2752 (44.5%)	10 (29.4%)	<0.05
Milk/Wool	74 (1.2%)	1 (2.9%)	
Mixed	1612 (26.1%)	13 (38.2%)	
Multi-production	234 (3.8%)	5 (14.7%)	
Self-consumption	1513 (24.5%)	5 (14.7%)	
**Presence of cattle in farm**			
No	3724 (60.2%)	22 (64.7%)	0.726
Yes	2461 (39.8%)	12 (35.3%)	
**Number of animals on the farm**			
Mean (SD)	38.7 (82.9)	193 (273)	<0.05
Median [Min, Max]	10.0 [0, 1250]	91.0 [0, 1140]	
**Pasture**			
No	6113 (98.8%)	34 (100%)	1
Yes	72 (1.2%)	0 (0%)	
**Movements**			
No	5352 (86.5%)	22 (64.7%)	<0.05
Yes	833 (13.5%)	12 (35.3%)	
**Number of Movements**			
Mean (SD)	0.189 (0.575)	0.706 (1.45)	<0.05
Median [Min, Max]	0 [0, 9.00]	0 [0, 7.00]	
**Suspension of the healthy status**			
No	6162 (99.6%)	9 (26.5%)	<0.05
Yes	23 (0.4%)	25 (73.5%)	

**Table 6 microorganisms-11-02623-t006:** Fisher and Wilcoxon tests and frequency table (2020).

	Negative	Positive	*p*-Value
	(N = 5718)	(N = 26)	
**Province**			
Avellino	1405 (24.6%)	1 (3.8%)	<0.05
Benevento	1134 (19.8%)	3 (11.5%)	
Caserta	626 (10.9%)	2 (7.7%)	
Napoli	390 (6.8%)	0 (0%)	
Salerno	2163 (37.8%)	20 (76.9%)	
**Productive orientation**			
Meat	2583 (45.2%)	8 (30.8%)	<0.05
Milk/Wool	62 (1.1%)	0 (0%)	
Mixed	1486 (26.0%)	12 (46.2%)	
Multi-production	206 (3.6%)	4 (15.4%)	
Self-consumption	1381 (24.2%)	2 (7.7%)	
**Presence of cattle in farm**			
No	5716 (100.0%)	26 (100%)	1
Yes	1 (0.0%)	0 (0%)	
Missing	1 (0.0%)	0 (0%)	
**Abortions**			
No	5712 (99.9%)	26 (100%)	1
Yes	6 (0.1%)	0 (0%)	
**Number of animals on the farm**			
Mean (SD)	40.0 (87.6)	182 (298)	<0.05
Median [Min, Max]	10.5 [0, 1580]	79.5 [7.00, 1420]	
**Pasture**			
No	5636 (98.6%)	24 (92.3%)	0.0548
Yes	82 (1.4%)	2 (7.7%)	
**Movements**			
No	4909 (85.9%)	22 (84.6%)	0.779
Yes	809 (14.1%)	4 (15.4%)	
**Number of Movements**			
Mean (SD)	0.205 (0.618)	0.423 (1.06)	0.673
Median [Min, Max]	0 [0, 11.0]	0 [0, 4.00]	
**Suspension of the healthy status**			
No	5689 (99.5%)	16 (61.5%)	<0.05
Yes	29 (0.5%)	10 (38.5%)	

**Table 7 microorganisms-11-02623-t007:** OR = Odds Ratio, CI = Confidence Interval 95%. * = iteration between the two variables.

Characteristic	OR	95% CI	*p*-Value
**years**			
2015	—	—	
2016	0.30	0.14, 0.63	0.002
2017	0.41	0.21, 0.80	0.009
2018	0.23	0.10, 0.53	<0.001
2019	0.05	0.01, 0.20	<0.001
2020	0.25	0.11, 0.57	0.001
**movements**			
No	—	—	
Yes	3.41	1.82, 6.41	<0.001
**suspension of healthy status**			
No	—	—	
Yes	55.8	26.7, 117	<0.001
**number of animals in herd**	1.04	1.03, 1.05	<0.001
**years * movements**			
2016 * 1	2.96	1.10, 7.97	0.031
2017 * 1	1.11	0.41, 3.01	0.8
2018 * 1	3.46	1.23, 9.74	0.019
2019 * 1	9.22	2.13, 40.0	0.003
2020 * 1	2.03	0.65, 6.28	0.2
**years * healthy status suspension**			
2016 * 1	0.92	0.31, 2.76	0.9
2017 * 1	1.91	0.63, 5.77	0.3
2018 * 1	5.55	1.60, 19.3	0.007
2019 * 1	34.6	8.04, 149	<0.001
2020 * 1	4.97	1.24, 19.9	0.023

## Data Availability

No new data were created or analyzed in this study. Data sharing is not applicable to this article.

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
