# Peer review of "Risk Factors for the Spread of Brucellosis in Sheep and Goats in the Campania Region in the Years 2015–2020"

_microorganisms, 2023, doi:10.3390/microorganisms11112623_

Round 1

Reviewer 1 Report

The paper "Risk factors for spread of Brucellosis in sheeps and goats in 2 Campania Region in the years 2015-2020" by Brunetti et all. talks about the risk factors of brucellosis in sheep and goat farms in Campania region in Italy. This research uses the multi-annual data in mixed effects logistics regression model in order to model year-to-year variations. The researchers combine the data from sheep and goat farms in a clever way to identify risk factors related to major factors known to contribute to spread of brucellosis in production animals. This is a very good exercise in use of science to improve veterinary management and cost effectiveness of incentivised measures not only regarding brucellosis, but other veterinary and public health diseases.

I have couple of major remarks:

1. Why haven't goat and sheep farms been separated and included as one of the risk factors?Especially because of the nature of two species, fodder used and variaty of breeds (that has not been mentioned at all in the paper).

2. The veterinary services in this paper appear to be mentioned only as official veterinary inspection. However, as a variable the veterinary services employed on farm hasn't been mentioned nor discussed. Neither did animal management routines, other than animal movement. For example: the existence of quarantine or pre-testing before introduction of new animals into the herd. Management of aborted foetuses and parturition materials, presence of shepherd dogs etc. For example echinococcosis was eradicated from Cyprus just by educating farmers and slaughter workers how to eliminate affected livers and intestines.

3. The presentation of statistical results is poorly executed. First two tables are unreadable with too much data. My suggestion would be to separate the presented data in several smaller tables and maybe keep these orginal ones as supplementary material.

Minor comments:

Literature should be updated and checked out since in several places either years or other information is missing and the literature list is not uniform.

Line 26 - Isn’t 55.8 major risk factor?

Line 35-38 - It might be useful to mention wildlife as one of the potential reservoirs for B. melitensis worldwide, and in this case

Line 47 - Is there more recent reference?

Line 52 - What are the foodstufs?

Line 88-90 - this is a repetition of the last sentences from previous paragraph

Line 98 - "... for the area". Suggestion: for the « given/selected/specific » area.

Line 106-108 - Not sure I understand this part of the statement. Please rephrase.

Line 133 - What does it mean "herds open at least one day..."

Line 137 - How many farms are there that haven’t been controlled at all in this period? Important for discussion not here, but omitted from discussion.

Line 154-155 - Repeated multiple times

Line 165-171 - In my opinion this is for discussion not M&M.

Line 186 - "companies" - Why use term company instead of farm?

Results in general:

There is no referencing to tables nor figures throughout this part of the paper. It seems that tables and figures exist separate from the rest of the manuscript.

Page 7 - "...the disease occurred more frequently in farms with a higher number of animals" - This has already been said on the previous page.

           - "...suspension of farm health status:..." - Was this suspension because of the brucellosis or any infectious disease or sanitary problem?

Page 9 - "... (Table 1 - OR = Odds Ratio, CI = Confidence Interval)" - This should be in the table legend not in the text?

Page 10 - "...indeed, both those that have introduced potentially infected animals from other farms, and those with a greater number of animals in the stable, have a higher risk of developing the disease over time." - it would be very interesting to see if the farms that have movements do testing and quaranteen prior to introduction of new animals. And even if the animals have been obtained outside the Campania region or within.

       - "Another aspect to consider is that not all pastures are georeferenced in Vetinfo and in any case during the processing phase the georeferencing included only two coordinates (latitude and longitude). " - Do two or more farms share pastures? In the regions of south-east Europe that is the most important risk factor for the transmission of brucellosis.

    -"Some territories, such as the prov- ince of Salerno most affected by the disease, are characterized by large expanses of land subject to transhumance of both cattle and sheep-goats: this favors direct and indirect con- tact of animals with the possible spread of infectious diseases." - In the same time, Salerno is in the middle of Campania region. Could it also be that farms from Salerno region mainly procure their animals from neighbouring provinces?

    - "...taking into account the widespread "transhumant" method of cattle breeding in this province which, in the past, caused one of the difficulties that hindered the containment of the in- fection in the area." - not sure I understand well this part of the sentence. Please rephrase.

Page 11. - "In fact, breeders often give little importance to this event by not communicating it to the Veterinary Services (S.V.) of the AA.SS.LL. I" - It would also be useful to see how many farms have ordinating veterinarian, who could in that case note those kinds of events or do farmers do everything by themselves.

   - "...induce..." - suggestion to change to urge/ incentivise

   - "The absence of a statistically significant association between abortions and the pres- ence of the infection could be attributable to the failure to notify them. In fact, breeders often give little importance to this event by not communicating it to the Veterinary Services (S.V.) of the AA.SS.LL. " - Already discussed above.

English language should be proof read, otherwise it is very good.

Author Response

Good morning, thank you for checking. Attached is the PDF with the response to the questions asked. We remain available for any further clarifications and/or suggestions.

Reviewer 2 Report

Dear Authors,

 Your study delves into a significant matter that holds a broad appeal to the readership – the assessment of potential risk factors affecting the proliferation of brucellosis within sheep and goat farms situated in the Campania region.

The objective of the study is indeed very important, and I am confident that the authors have invested considerable effort and dedication in its execution. However, there are several shortcomings in the writing and structure of the article. The methods need to be more comprehensively described to support the findings. The tables require improvements in various aspects. Furthermore, there are numerous shortcomings in the references.

Unfortunately, in my opinion, the manuscript needs significant refinement to be considered for publication. Attached is the manuscript with suggested revisions for your consideration.

The quality of the English needs improvements.

Author Response

Thanks so much for the suggestions; we have made the requested corrections. We remain at your disposal for any further clarifications, changes and/or additions to be made to the manuscript. Were the changes made directly to the work: should we upload it to the platform so that all reviewers, besides you, can verify the changes? 

Reviewer 3 Report

The article by Ottaiano et al. is interesting, scientifically recognized, relevant to the scope of the journal.  The article dealing with risk factors for spread of Brucellosis in sheeps and goats in part of Italy and providing useful information about the possible risk factors influencing the spread of brucellosis among sheep and goat farms in the Campania region. Despite the study being well designed, however, the are few requirements that are needed to improve the text. Few comments and corrections are required before publication as the following:

·         Abstract is missing a recommendation

·         B. melitensis can also infect other animals, please use new and updated refernces Doi: https://doi.org/10.51585/gjvr.2022.1.0028

·         Ensure that all names of bacteria in italic, as in lines 41, 43, etc.

·         Line 55, the ingestion of contaminated products of animal origin, please give an example doi: 10.1111/tbed.12535

·         Lines 76-81, authors used very old refernces [Kabagambe et al., 2001; Kadohira, 1997; Omer, 2000], please replace by recent  and updated refernces. For example Doi: https://doi.org/10.51585/gjvr.2022.1.0033 and https://link.springer.com/chapter/10.1007/978-981-13-8844-6_10

·         Line 96: (O.M. may 2015 ss.mm.ii). ????

·         The second half of the discussion is missing support by relevant refernces.

Author Response

Thanks so much for the suggestions; we have inserted the indicated references. We are waiting to know whether to upload the edited manuscript back to the platform so that all reviewers can view it.

Round 2

Reviewer 2 Report

Dear authors,

I can see that you are making improvements to the article based on the suggestions provided. However, I still notice that the manuscript has not reached the level of quality required for publication. I strongly recommend that you seek the services of a professional proofreader before making a new submission.

Sincerely

Must be reviewed by a proof reader.

Author Response

Good morning, we have had the English form of the work corrected by a professional from our Institute

Thank you soo much

Reviewer 3 Report

The authors added the suggested reference in the place prepared for the citation of this article. You have to improve the introduction by including information in this article. 

Author Response

Good morning, we have made the requested additions

Thank you soo much